# Social support and professional networks of nurses and nursing technicians in coping with Covid-19: A sectional study in two Brazilian cities

Helena Maria Scherloski Leal David[1]*, Maria Rocineide Ferreira da Silva[2‡], Magda Guimarães de Araújo Faria[1‡], Tarciso Feijó da Silva[3‡], Tatiana Cabral da Silva Ramos[4‡], Marcus Vinicius Pereira-Silva[5‡]

1 Public Health Nursing Department, State University of Rio de Janeiro, Rio de Janeiro, Rio de Janeiro, Brazil, 2 Health Sciences Center, State University of Ceará, Fortaleza, Ceará, Brazil, 3 Health Sciences Center, Federal University of Pará, Belém, Pará, Brazil, 4 Nursing Department, Estacio de Sá University, Duque de Caxias, Rio de Janeiro, Brazil, 5 Casa de Oswaldo Cruz, Oswaldo Cruz Foundation, Rio de Janeiro, Rio de Janeiro, Brazil

‡ MRFS and MGAF are joint senior authors. TFS, TCSR, and MVPS are assistant authors and methodological reviewers.
* helenalealdavid@gmail.com

## Abstract

Among healthcare workers, nurses are at exceptionally considerable risk for contracting COVID-19. Regardless of professionals' level of education, Brazilian nursing is one of the healthcare occupations shouldering the highest levels of responsibility and workload. Social support networks to health and nursing can be a strategy to reduce workload and stress and may contribute to implementing the activities and protecting workers' health. This study aimed to map and analyze social support networks at workplaces as informed by frontline nursing professionals working in healthcare units in the Brazilian cities of Rio de Janeiro and Fortaleza, capitals of the States of Rio de Janeiro and Ceará, respectively. This observational and cross-sectional study used an online data collection instrument based on social network analysis methodology. We recruited 163 participants in two reference services for health professionals suspected or with COVID-19 symptoms. The research question was: "Which category or categories of health professionals have supported you or other colleagues the most in the event of diagnosis or suspicion of COVID-19 among nursing categories? Data were organized by nursing category and city and analyzed through social network analysis using Ucinet©, generating graphs and centrality metrics. Results point to the central relevance of nursing categories in the workplace social support in the pandemic, followed by other health professional categories.

## Introduction

The nursing staff is among the most vulnerable occupational groups to control infectious diseases, which is mainly because these professionals work intensively in providing direct care to

**Data Availability Statement:** All relevant data are within the paper and its Supporting Information files.

**Funding:** The author(s) received no specific funding for this work.

**Competing interests:** The authors have declared that no competing interests exist.

patients, thus putting their lives at risk and increasing the pressure exerted on their work process [1].

In Brazil, the regulations that define the set of actions and resources for protecting healthcare and nursing professionals are found in the Regulatory Standard 32 [2]. However, the epidemic control measures established by the World Health Organization [3], which include protective strategies and environmental and administrative actions [4], are sometimes not complied with. In the Brazilian territory, they are still primarily focused on using Personal Protective Equipment [5].

There are two professional Brazilian nursing categories, by work process and education: nurses and nursing technicians. The first are higher education professionals responsible for organizing, managing, and coordinating nursing services [6]. The nursing technicians are professionals with a high-school education, and their training takes, on average, 18 months. Supervision and coordination are conducted by nurses, and practical tasks are performed by technicians and assistants (although nurses should assume direct care for the most severely ill patients). In general, the nursing workforce is mainly provided by women who seek education and enter the labor market. A quick technical school preparation is extremely attractive and does not require the time needed for a university degree. Nursing technician wages are significantly lower than that of professional 'Nurses' [7].

Regardless of a professional's level of education, Nursing is one of the healthcare occupations shouldering the highest levels of responsibility and workload, which directly influences the quality of the care provided [8]. One of the strategies used by health professionals to reduce workload and stress is developing a social support network at work, which is defined as the existing interaction between individuals involved in the work process (including employees, supervisors, and managers), built cooperatively and contributing to implementing the activities developed and protecting workers' health [9].

Different arrangements and strategies to compose nursing teams in Brazil may be employed in the daily health care workplace, especially when it is necessary to provide care in situations of adversity and scarcity of resources [7]. In the COVID-19 pandemic, nursing professionals had to provide effective care to secure specialized and high-quality care per patient needs. Also, the category is engaged in the containment of outbreaks, the establishment of care protocols, the provision of materials and protective equipment, clinical management, and the provision of professional qualification for the health team [10].

The nursing work dynamics and the interactions between health team actors during the working day are not immediately visible. The Social Network Analysis (SNA) methodology can enlighten formal and informal relationships in the context of nurse and nurse technicians' search for social support, which includes investigating how nursing professionals have been facing the daily risks of contracting COVID-19, and how social support is produced in the workplace to respond to situations of suspected or confirmed COVID-19 infection of nursing personnel.

In June 2020, the State of Rio de Janeiro had 185,790 registered nursing technicians and 55,413 nurses. The State of Ceará had 42,186 nursing technicians and 20,831 nurses [11]. Concerning COVID-19 suspected or confirmed cases and deaths, the Nursing Observatory of the Brazilian Federal Council of Nursing indicates that, on June 23, 2020, 3,877 cases had been recorded in Rio de Janeiro, with 32 deaths, and 1,352 cases in Ceará, with 11 deaths.

Currently, almost two years into the pandemic, protocols and therapeutic procedures are already well established all over Brazilian healthcare services. However, during this research, there was no broad and immediate access to diagnostic tests and medical follow-up neither for the population nor for health professionals with suspected COVID-19. We can, thus, admit

that social support networks of health workers were produced or strengthened (if they already existed) during this initial phase of the pandemic. We could not find any Brazilian study about social support networks and nursing work related to this pandemic.

This study aimed to map and analyze the existing social support networks in the work of nursing professionals who became ill or were suspected of being infected by COVID-19 and went to emergency services of reference for health professionals in Rio de Janeiro, capital of the State of Rio de Janeiro, and Fortaleza, capital of the State of Ceará, during this initial period of the Brazilian pandemic.

## Social networks, nursing work, and social support

Social networks can be defined as a set of relationships between different social entities (also called actors), such as people, organizations, documents, and groups. These relationships could be based on solidarity or shared interests and objectives, with mobilization of the actors through interactions and continuous sharing of information and resources. These relationships represent a social fabric that can strengthen social actors around common objectives, including facing collective problems [12].

The Social Network Analysis (SNA) was developed as a research methodology from the 1930s onwards, within social sciences and psychology. It can today be supported by computerization processes, which expand the possibilities of calculations based on the relationships between the actors of a network and allow visualization through sociograms, which are visual representations based on the mathematical relationships between actors (also called vertices and nodes) and links (also called edge or connections). In a network, questions and data are presumed to be organized to ensure that each link in a sociogram means the same thing. Each link represents the same type of social relationship [13].

Regarding the characterization of SNA, one can study a network from its minimum configuration, the dyad or link between two actors, and the attributes of the network as a whole, in its full breadth and complexity.

Social networks can be categorized by the diversity of social entities or actors. Networks with only one set of actors are called 1-mode networks. On the other hand, networks with two sets of actors or one set of actors and events are called 2-mode networks. The SNA can also have an ego-centered perspective. In these cases, the analyses are concentrated on a particular group of respondents and the ties that this group has to others. This perspective is frequently used in social support studies [14].

In this study, the actors are nurses and nursing technicians, and the professional positions mentioned are considered another set of actors. Since SNA does not concern itself with personal and individual attributes but the relationships between actors and alters, we have chosen to study these networks as 2-mode networks and an ego-centered perspective. The respondent actors (nursing technicians or nurses) indicate which team members (jobs or positions) stand out, in their experience, as relevant in the informal social support network for workers with diagnosis or suspicion of COVID-19.

For SNA purposes, this type of network is considered to present structural equivalence: nursing professionals are structurally equivalent since they connect with the same set of professional categories existing in the services (alters) [15]. The analysis of 2-mode networks generate centrality measures, where the most central alters are those most frequently mentioned or stand out for each actor interviewed. These central alters are not all necessarily part of the same total network and do not necessarily know each other [16].

The social support concept is widely disseminated and studied by the scientific community. It strengthens existing relationships in each network, whether through trust or mutual help

among those involved. In crises, social support allows, encourages, and (re)builds networks, thus contributing to individual and collective well-being [17].

Social support at work is defined as assistance and protection provided to professionals by colleagues, leaders or managers, individually or collectively, and is hypothesized to be reciprocal [18]. Several studies discuss the relationship between social support and reduced stressful situations, integration with social groups, increased satisfaction and sense of belonging, and, not least, mutual help relationships [19]. This support is very crucial for nurses to address and control different stressors in their work environment.

Specifically, social support at work has been related to job satisfaction and even with the permanence of individuals in their jobs. The most evident form of social support at work helps perform tasks or solve problems [20].

Social support at work is crucial for developing care activities in nursing, particularly those performed in multidisciplinary fashion. It is noteworthy that these relationships are a protective factor regarding health, especially mental health [21].

The role of nursing professionals as leaders and providers of social support at work is emphasized, as they facilitate the development of strategies such as hope, self-efficacy, resilience, and optimism, especially in crises such as the current pandemic [22, 23].

Specifically, in the COVID-19 pandemic backdrop, social support at work is seen as an effective tool in maintaining the quality of life and coping with events that may affect the quality of the care provided. It, therefore, is a device to impel resilient practices [24].

Nursing work in the COVID-19 pandemic has particularities related to social support, especially with the difficulties related to the work process. Despite the intense distress when addressing death and issues of personal illness, nurses see themselves as part of a relentless care organization, generating contradictory sensations of obligation and satisfaction, besides surfacing the need for continuing education in health, where learning with the team members themselves becomes the most recurrent and viable strategy, which points to the need for a social network established among the team [25].

## Materials and methods

This observational and cross-sectional study is based on the social network analysis methodology based on 2-mode and ego-centered perspective, using an online data collection platform.

### Network data and collection

The instrument included sociodemographic and professional variables, exploratory questions on PPI training, and two SNA questions about social support and access to PPI. The central question to nurses and nursing technicians was: Which category or categories of health professionals have supported you or other colleagues the most in the event of diagnosis or suspicion of COVID-19 among health categories?

Respondents answered by selecting up to five professional categories from the list provided containing the names of jobs or positions that usually make up the service or support teams that work with COVID-19 cases: nurse, nursing assistant or technician, infectious disease physician, nursing director, physiotherapist, general practitioner, laboratory assistant or technician, pulmonologist, doctors in other specialties, multidisciplinary training team, pharmacy assistant or technician, general manager, driver, administrative assistant, social worker, nutritionist, community health agent, warehouse team, pharmacist, municipal or state manager, and biologist.

There was no previous test of validity of the questionnaire, but it was evaluated by six nurses who worked in the services regarding the understanding of the questions.

## Setting

We recruited participants by purposeful sampling in two reference institutions that centralized healthcare for health professionals with diagnosis or suspicion of COVID-19 in Rio de Janeiro and Fortaleza from June 23 to August 5, 2020. In Rio de Janeiro, the unit where the survey was conducted was a large outpatient service managed by a public university, while in Fortaleza, it was a municipal hospital emergency center designated to receive suspected or sick professionals. The availability of these services to all health professionals was widely publicized.

## Sample

Participants were personally invited and asked to answer the questionnaire when they sought care at the institutions, and the inclusion criteria were adults, acceptance of participation, being a nursing professional working in direct contact with the population at large or with people hospitalized with COVID-19 and who have been suspected or infected by COVID-19 while working. Participants worked in different public and private hospitals and services in each city. They were recruited among nursing professionals that could assign for testing and diagnosis at the reference services, and the study did not include those hospitalized now of data collection.

We had 99 respondents from Rio de Janeiro (56 nursing technicians and 43 nurses) and 64 from Ceará (12 nursing technicians and 52 nurses). The sample size was not previously defined. Temporal criterion was adopted to end the data collection, which was carried out in the months of greatest demand for testing or treatment by health professionals.

The answers were obtained after the informed consent form was signed and stored in an electronic spreadsheet database. They were later processed to generate the mode-2 matrix, where the lines contained the responses of the nursing professionals by category, and the columns showed the professional categories previously listed.

Research design and development can be synthesized in the following scheme (Fig 1).

## Mathematical representation of social network

The centrality metrics were: degree centrality, intermediation centrality, and proximity centrality.

| Research Project Formulation | Data colletion and processing | Data analysis (SNA) |
|---|---|---|
| **Methodology:** 2-mode and ego-centered network perspective<br><br>**Actors:** nursing professionals<br>• Nurses<br>• Nursing Technicians<br><br>**Question:** Which category or categories of health professionals have supported you or other colleagues the most in the event of diagnosis or suspicion of Covid-19 between health categories?<br><br>**Alters:** services jobs or positions<br>• Administrator<br>• Chief nurse<br>• Team nurse<br>• Nursing assistant or technician<br>• General physician<br>• Assistant or laboratory technician<br>• Infectologist doctor<br>• Doctors with other specialties<br>• Assistant or pharmacy technician<br>• Administrative assistant<br>• Multiprofessional training team<br>• Pharmacist<br>• Social assistant<br>• Physical therapist<br>• Nutritionist<br>• Community health agent<br>• Driver<br>• Municipal or State manager<br>• Biologist | **Sceneries:** Health services that where designed to be reference to all health professionals with Covid-19 suspicion<br><br>**Cities:** Rio de Janeiro (Rio de Janeiro) and Fortaleza (Ceará)<br><br>**Total of respondents:** n=164<br>Exclusions: n= 4 (Nursing assistants)<br>Rio de Janeiro: n= 99 (43 nurses, 56 nursing technicians)<br>Fortaleza: n= 64 (52 nurses, 12 nursing technicians)<br><br>**Period:** June 23rd to August 5th, 2020<br><br>**Instrument:** Online Google Forms questionnaire, generating an Excel© spreadsheet<br><br>**Processing:**<br>• 1st step: spreadsheet review and anonymization<br>• 2nd step: Excel© matrix selecting actors, nursing categories, cities and mentioned alters<br>• 3rd step: exportation to UCINET© for metric analysis<br>• 4th step: exportation to Gephi© for social network visualization (sociogram)<br><br>**Ethical requirements:** Submitted to National Research Ethics System. Approval document number: 4094637 | **Softwares:**<br>• Excel©<br>• Ucinet©<br>• Gephi©<br><br>**Network category:** 2-mode<br><br>**Metrics:**<br><br>• Degree centrality: number of connections<br><br>• Closeness centrality: distance of a given actor in relation to other actors in the network<br><br>• Betweenness centrality: related to actors' ability to mediate information and actions in the network |

**Fig 1. Research design and development scheme.**

Degree centrality (degree) is related to the actor's number of connections. In this research, the mode-2 network is based on a bipartite graph. All paths in this type of graph consist of an alternating series of nodes and edges (actors and connections), with two sets of actors or vertices. In bipartite graphs used in the analysis of mode-2 networks, degree centrality is determined by the number of connections originated from actors of another category or entity [15, 16], meaning that in the analysis of social support networks in nursing work related to coping with COVID-19, each professional category's degree centrality was determined by the number of times nurses and technicians cited it. Therefore, the actors with the highest degree of centrality were professionals most often indicated by nursing professionals as the categories who most supported them in coping with COVID-19.

The centrality of proximity (closeness) is related to the distance of a given actor from other actors in the network. Therefore, actors with greater proximity centrality are closer to the other actors in the network. In bipartite graphs, the minimum distance between one actor and another in the same set is 2, and the minimum distance between two actors in different sets is 1 [15]. In social support networks, the actors with greater proximity centrality are closer to the other actors, thus providing support quicker.

The centrality of intermediation (betweenness) is related to actors' ability to mediate information and actions in the network. Actors with higher centrality of intermediation have greater power to mediate information and network actions to support the different actors in the network. Calculation of the intermediation centrality measure is based on the geodesic paths, i.e., the shortest paths between two actors. In bipartite graphs, paths can start and end in either of the two sets of actors [16].

The matrices were analyzed using UCINET© software, and centrality measures were generated. Subsequently, data was migrated to Gephi© software to obtain sociograms, which were organized according to the categories of nursing technicians and nurses in each city, generating four sociograms (Fig 2). We also included centrality grade tables containing both nursing categories measures (Tables 1 and 2).

This investigation is part of a larger study that analyzed also the social networks related to personal protective equipment-PPE access by nursing personnel, and was previously assessed and approved by the Human Research Ethics Committee of the Brazilian National Research Ethics System.

## Results

Distribution ranking of each measure of centrality between the two categories of nursing professionals (nurses and nursing technicians) are summarized in Tables 1 and 2:

Nurses were the most frequently recognized professionals for providing social support to the respondent nurses and technicians, followed by nursing technicians, general practitioners, laboratory assistants, and physiotherapists. These results were similar in the networks of nurses and technicians in both states where the research was conducted.

However, the results related to the proximity measure show differences between the two states, both in the total of measures and in the actors that stand out. For nurses in Rio de Janeiro, besides nurses, nursing technicians, general practitioners, laboratory technicians, and physiotherapists standing out, the multidisciplinary training team also obtained a significant score. The general management, the multidisciplinary training team, and doctors in other specialties obtained the same score for nurses in Fortaleza. The intermediation score is higher for nurses and nursing technicians in both states. The mediating role played by these two actors stands out compared to the other categories.

## Nurses' social support network

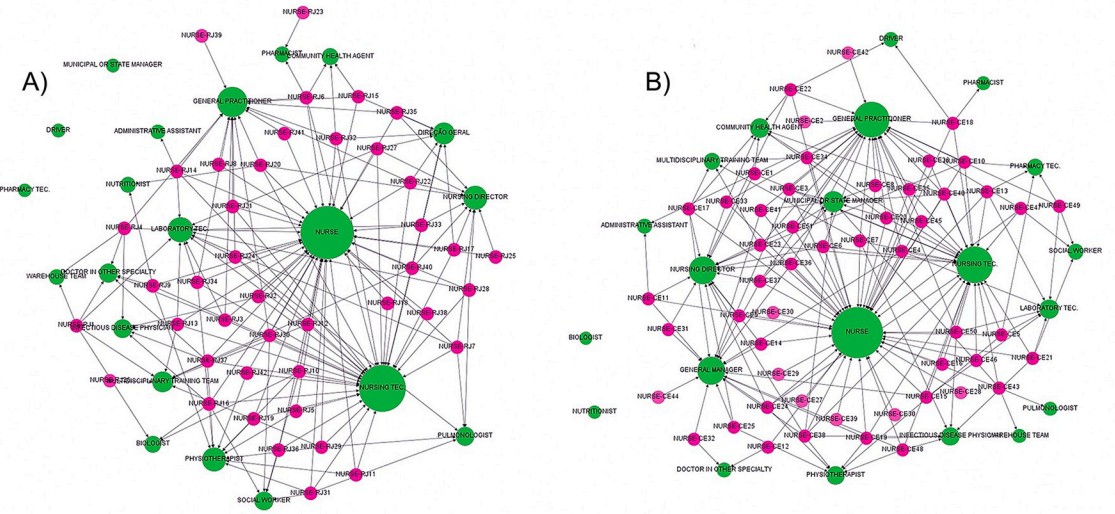

## Nurses' technicians'social support network

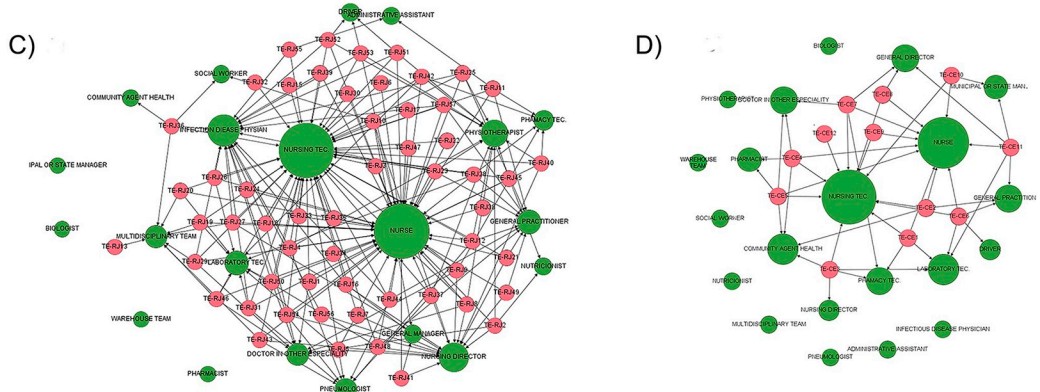

**Fig 2. Nurses and nursing technicians'social support network sociograms–Rio de Janeiro and Ceará.** (A) Rio de Janeiro Nurse´s Sociogram. (B) CEará Nurse´s Sociogram. (C) Rio de Janeiro Nursing Technician´s Sociogram. (D) Ceará Nursing Technician´s Sociogram.

Nursing technicians' responses in Rio de Janeiro and Fortaleza evidence that nurses and nursing technicians are the most essential categories regarding centrality degree. Other professionals have way lower centrality degrees than these two nursing categories despite being mentioned. It is worth noting that some categories were not even mentioned, such as drivers and state managers.

## Social support sociograms

In sociograms, the respondents (actors) are represented by codes in red circles, and green circles indicate the professionals or positions mentioned. The size of the circles is proportional to the centrality degree of the link (Fig 2, Sociograms A, B,C and D)

Professional colleagues stand out again in both networks of nurses, but it is interesting to note that general practitioners appear to be more prominent here than in the networks of nursing technicians (Sociograms A and B).

**Table 1. Centrality measures of nurses' social network–Rio de Janeiro and Ceará.**

| | Rio de Janeiro | | | Fortaleza | | |
|---|---|---|---|---|---|---|
| | **Degree** | **Closeness** | **Betweenness** | **Degree** | **Closeness** | **Betweenness** |
| Nurse | 0.905 | 0.976 | 0.394 | 0.846 | 0.885 | 0.475 |
| Nursing technician or assistant | 0.762 | 0.837 | 0.237 | 0.519 | 0.657 | 0.135 |
| General practitioner | 0.381 | 0.631 | 0.087 | 0.500 | 0.622 | 0.150 |
| Nursing director | 0.238 | 0.532 | 0.014 | 0.327 | 0.541 | 0.059 |
| General manager | 0.190 | 0.494 | 0.008 | 0.327 | 0.541 | 0.059 |
| Municipal or state manager | 0.000 | 0.000 | 0.000 | 0.173 | 0.495 | 0.013 |
| Laboratory assistant or technician | 0.262 | 0.554 | 0.033 | 0.135 | 0.469 | 0.008 |
| Community health agent | 0.071 | 0.451 | 0.001 | 0.135 | 0.469 | 0.006 |
| Physiotherapist | 0.238 | 0.512 | 0.014 | 0.115 | 0.469 | 0.004 |
| Infectious disease physician | 0.119 | 0.494 | 0.005 | 0.115 | 0.451 | 0.006 |
| Multidisciplinary training team | 0.167 | 0.526 | 0.042 | 0.077 | 0.447 | 0.001 |
| Pharmacy assistant or technician | 0.000 | 0.000 | 0.000 | 0.077 | 0.442 | 0.003 |
| Social worker | 0.119 | 0.482 | 0.003 | 0.058 | 0.422 | 0.001 |
| Doctor in other specialty | 0.119 | 0.494 | 0.009 | 0.038 | 0.404 | 0.001 |
| Administrative assistant | 0.024 | 0.436 | 0.000 | 0.038 | 0.434 | 0.000 |
| Driver | 0.000 | 0.000 | 0.000 | 0.038 | 0.426 | 0.000 |
| Pulmonologist | 0.119 | 0.482 | 0.003 | 0.019 | 0.397 | 0.000 |
| Pharmacist | 0.048 | 0.441 | 0.031 | 0.019 | 0.407 | 0.000 |
| Warehouse team | 0.048 | 0.461 | 0.001 | 0.019 | 0.397 | 0.000 |
| Biologist | 0.071 | 0.471 | 0.004 | 0.000 | 0.000 | 0.000 |
| Nutritionist | 0.048 | 0.451 | 0.000 | 0.000 | 0.000 | 0.000 |

The centrality degree of nursing technicians in Rio de Janeiro shows that they emphasize the support of nurses and their colleagues more than that of other professionals and managers (general management and nursing management). Staff members who are heads of hierarchical structures (nursing director, hospital chief) are less frequently mentioned but still appear in the network (Sociogram C)

Nurses and nursing technicians also stand out in the network of nursing technicians in Fortaleza. In this network, two healthcare categories were not mentioned in the Rio de Janeiro network: the community health worker (ACS) and the municipal secretary of health/health manager. Many professionals and teams were not even mentioned, and the number of nursing technician respondents was lower than that in Rio de Janeiro (Sociogram D).

Some teams, positions, or professionals were not mentioned by any or hardly any of the respondents. Biologists, nutritionists, drivers, physiotherapists, social workers, medical specialists, and laboratory assistants are rarely mentioned.

## Discussion

The contexts of the two cities surveyed are different. Although both are state capitals, Rio de Janeiro has a more significant offer of healthcare services and professionals. Even with these differences in the local systems, the networks obtained were similar between the two categories studied (nurses and technicians).

The highest degrees obtained in the centrality metrics (Table 1) are all related to the nurse and the nursing technician categories. Technicians are the most numerous nursing category in the country and correspond to 70% of the nursing workforce [11].

**Table 2. Centrality measures of nursing technicians' social network–Rio de Janeiro and Ceará.**

| | Rio de Janeiro | | | Fortaleza | | |
|---|---|---|---|---|---|---|
| | **Degree** | **Closeness** | **Betweenness** | **Degree** | **Closeness** | **Betweenness** |
| Nursing technician or assistant | 0.875 | 0.941 | 0.355 | 0.917 | 1.444 | 0.191 |
| Nurse | 0.875 | 0.923 | 0.335 | 0.833 | 1.300 | 0.134 |
| Infectious disease physician | 0.357 | 0.585 | 0.055 | 0.000 | 0.000 | 0.000 |
| Nursing director | 0.304 | 0.558 | 0.042 | 0.083 | 0.743 | 0.000 |
| Physiotherapist | 0.250 | 0.522 | 0.017 | 0.000 | 0.000 | 0.000 |
| Laboratory assistant or technician | 0.179 | 0.500 | 0.008 | 0.333 | 0.963 | 0.028 |
| General practitioner | 0.179 | 0.490 | 0.008 | 0.250 | 0.867 | 0.008 |
| Doctor in other specialty | 0.143 | 0.485 | 0.006 | 0.250 | 0.839 | 0.005 |
| Multidisciplinary training team | 0.143 | 0.485 | 0.030 | 0.000 | 0.000 | 0.000 |
| Pulmonologist | 0.143 | 0.480 | 0.004 | 0.000 | 0.000 | 0.000 |
| Pharmacy assistant or technician | 0.089 | 0.462 | 0.002 | 0.250 | 0.867 | 0.008 |
| Social worker | 0.036 | 0.449 | 0.001 | 0.000 | 0.000 | 0.000 |
| General manager | 0.054 | 0.440 | 0.001 | 0.250 | 0.867 | 0.008 |
| Driver | 0.054 | 0.436 | 0.000 | 0.083 | 0.743 | 0.000 |
| Administrative assistant | 0.036 | 0.436 | 0.000 | 0.000 | 0.000 | 0.000 |
| Nutritionist | 0.036 | 0.436 | 0.000 | 0.000 | 0.000 | 0.000 |
| Community health agent | 0.018 | 0.414 | 0.000 | 0.333 | 0.929 | 0.019 |
| Pharmacist | 0.000 | 0.000 | 0.000 | 0.167 | 0.765 | 0.000 |
| Municipal or state manager | 0.000 | 0.000 | 0.000 | 0.167 | 0.788 | 0.002 |
| Warehouse team | 0.000 | 0.000 | 0.000 | 0.000 | 0.000 | 0.000 |
| Biologist | 0.000 | 0.000 | 0.000 | 0.000 | 0.000 | 0.000 |

Social support in nursing work provided by coworkers in the same professional category has been continuously highlighted in Brazilian scientific literature. Nurses are uniquely relevant in maintaining social relationships at work, which can be associated with what could be called *prosperity at work* [25]. This situation occurs mainly due to the importance of nursing professionals in problem-solving and care continuity.

Research carried out in recent years in Brazil has identified support networks for people with tuberculosis and Hansen's disease. Such networks focus on coping with the disease [26, 27] and monitoring tuberculosis cases [28]. The studies have also analyzed the influence of social networks on prenatal care and care provided to hypertensive and diabetic patients [29–32] and have helped understand how social relationships influence health care management and regulation [33, 34]. Nurses have consistently emerged as central actors in providing care in all these studies, especially in Primary Care, as shown in another study focused on service reception teams in Rio de Janeiro City [35].

International studies adopting the SNA methodology with healthcare professionals show that nurses have higher centrality degrees compared to other actors. A study conducted with professionals working in the intensive care unit of a university hospital in Japan found that the interactions between nursing professionals were significantly more active than those of other professionals. Moreover, nurses were also the category that interacted the most with other categories [36]. A similar result was found in another investigation in Japan, where the structure of a community clinical service network indicated nurses as the central actor in the social network [37]. A Canadian analysis revealed that aspects such as cohesion and communication between four teams were positive for shared decision-making ability, and nurses play an essential role in a non-formal hierarchy [38].

Such results endorse the role of nursing mediation and communication within the teams, which can also be related to professional training to develop empathy, resource management, and listening capacity [39]. On the other hand, it should be noted that the structure of a health service's social network does not follow any type of predetermined or formal hierarchy, and is based solely on the relational interaction between its members, meaning that the central actors are not necessarily the formal leaders of the network [40].

In this sense, the identification of the nurse category as a central actor points to the category's connection attributes and the fact that this actor's participation is directly related to their leadership capacity, which is based above all on cultural competence and ability to communicate and generate trust related to contextual and individual factors [41, 42].

Three characteristics are inherent to the actors in a network that influence the role assumed in the social network, particularly the central roles in the professional networks: 1) understanding characteristics of the coworker's work process; 2) valuing the coworker's knowledge on a specific subject, particularly one related to the problem to be solved in the work environment; 3) allowing access to one's ideas, earning a place for oneself in the role of network collaborator [43].

Removing a nurse with a leading role in a network can generate significant repercussions for all the other actors [44], as informed in a study where a nurse with a high degree of centrality participated in a healthcare professionals' network that cared individuals infected with Ebola. The event of her death caused great anxiety in the other workers and loss of confidence in health service administrators, who could be considered the formal leaders in that workplace.

When the informal leadership and the high centrality degree of professional nurses are observed, their responsibility within the social networks for fighting and coping with COVID-19 and their role in keeping themselves healthy becomes evident, since a possible illness can influence the entire network.

Nursing technicians also stand out as central mediators in the networks, which indicates that their performance exceeds the merely technical dimension of the work. As this category has more significant numbers than registered nurses, these professionals are expected to act with a spirit of leadership, although their direct relationships with medical professionals are less evident than those of nurses.

The fact that other nursing professionals are highlighted cannot be considered a result free from bias due to professional identity. On the other hand, the starting point to identify the network was a question addressed to each nursing actor (egocentric network) in a separate research set so that other professionals and categories could be chosen as well.

The differences in both categories' direct relational ties to doctors also raise hypotheses regarding the reproduction of class, gender and race relationships at work since Brazilian nursing is exercised mainly by women. Among nursing technicians, more than half are black women [44].

Participants were nursing professionals that could attend the referral service and excluded those that developed rapid and severe symptoms to the point of being immediately hospitalized, which is a limitation of this study.

## Conclusion

Informal social networks in the workplace are produced in adversity, scarcity of resources, and social and health system disorganization, as observed during the outbreak of the COVID-19 pandemic, and are homogeneous when considering the statements of nursing professionals. However, we identified differences between the two categories (nurses and nursing assistants or technicians).

Nursing staff relied on each other for informal yet resolutive support in emergency needs related to facing the threats posed by the pandemic to these workers. The SNA methodology helped to evidence the relational ties between nursing professionals and other professionals, including those in the same category.

## Supporting information

**S1 File.**
(DOCX)

**S1 Data.**
(CSV)

**S2 Data.**
(CSV)

**S3 Data.**
(CSV)

**S4 Data.**
(CSV)

## Acknowledgments

The authors are grateful to all the nurses and nursing technicians who participated in this study.

## Author Contributions

**Conceptualization:** Helena Maria Scherloski Leal David.

**Data curation:** Helena Maria Scherloski Leal David.

**Formal analysis:** Helena Maria Scherloski Leal David, Magda Guimarães de Araújo Faria, Tarciso Feijó da Silva.

**Investigation:** Maria Rocineide Ferreira da Silva, Tarciso Feijó da Silva, Tatiana Cabral da Silva Ramos.

**Methodology:** Helena Maria Scherloski Leal David, Maria Rocineide Ferreira da Silva, Marcus Vinicius Pereira-Silva.

**Project administration:** Helena Maria Scherloski Leal David, Magda Guimarães de Araújo Faria.

**Resources:** Tarciso Feijó da Silva, Tatiana Cabral da Silva Ramos.

**Software:** Marcus Vinicius Pereira-Silva.

**Supervision:** Helena Maria Scherloski Leal David, Maria Rocineide Ferreira da Silva.

**Validation:** Maria Rocineide Ferreira da Silva, Magda Guimarães de Araújo Faria, Marcus Vinicius Pereira-Silva.

**Visualization:** Marcus Vinicius Pereira-Silva.

**Writing – original draft:** Helena Maria Scherloski Leal David, Maria Rocineide Ferreira da Silva, Magda Guimarães de Araújo Faria, Tarciso Feijó da Silva, Tatiana Cabral da Silva Ramos, Marcus Vinicius Pereira-Silva.

**Writing – review & editing:** Helena Maria Scherloski Leal David.

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
