## [Decision Letter · Decision Letter 0]

25 Jan 2022

PONE-D-21-24493Social support and professional networks of nurses and nursing technicians in coping with Covid-19: a sectional study in two Brazilian citiesPLOS ONE

Dear Dr. David,

Thank you for submitting your manuscript to PLOS ONE. After careful consideration, we feel that it has merit but does not fully meet PLOS ONE’s publication criteria as it currently stands. Therefore, we invite you to submit a revised version of the manuscript that addresses the points raised during the review process.

We look forward to receiving your revised manuscript.

Kind regards,

José S. Andrade Jr.

Academic Editor

PLOS ONE

https://journals.plos.org/plosone/s/file?id=ba62/PLOSOne_formatting_sample_title_authors_affiliations.pdf”.

We note that you have reported significance probabilities of 0 in places. Since p=0 is not strictly possible, please correct this to a more appropriate limit, eg 'p<0.0001'.

A clean copy of the edited manuscript (uploaded as the new *manuscript* file).

Reviewers' comments:

Reviewer's Responses to Questions

**Comments to the Author**

1. Is the manuscript technically sound, and do the data support the conclusions?

Reviewer #1: Partly

Reviewer #2: Partly

2. Has the statistical analysis been performed appropriately and rigorously? 

Reviewer #1: No

Reviewer #2: No

3. Have the authors made all data underlying the findings in their manuscript fully available?

Reviewer #1: No

Reviewer #2: Yes

4. Is the manuscript presented in an intelligible fashion and written in standard English?

Reviewer #1: Yes

Reviewer #2: No

5. Review Comments to the Author

Reviewer #1: I have appreciated the manuscript and I think the following points must be clarified/improved:

1. The authors could double check the structural and grammar aspects in the following sentences:

“Networks can, in terms of homogeneity and reciprocity between their actors, be classified as type-1 (total network) or mode-2.”

and

“It’s an observational and cross-sectional research, based on the social network analysis methodology, using an online data collection instrument.”

2. The authors state that

“The study of social networks began in the 1930s within the scope of the social sciences.”

and

“The concept of social support is widely disseminated and studied by the scientific community.”

These statements should be referenced.

3. The authors should be clear to all readers, not only those from Social Science. Therefore, I suggest that they introduce the nomenclature used by social scientists, physicists, and mathematicians in the studying of social networks at the beginning and adopt only one throughout the manuscript. For example, the authors state that

“The former exists when the actors (nodes) who mention other actors (also called links or alters) are part of the same network, in a square matrix.”

Links are edges (ties) for Physicists. The authors should avoid this kind of problem.

4. It is imperative that the authors include a didactic figure explaining their model precisely. Further, I think the authors could emphasize that their model is a bipartite network at the beginning, not only at the ending.

5. I think the Social Network Analysis performed was quite superficial. There are no histograms or scatter plots of any kind, there is only 1 table and 4 figures (which could be combined into a single panel). Further, the quality of all figures is very low. The authors should improve their analysis.

6. Why are the results not biased due to the chosen sample of participants (only nurses and nursing technicians)? For example, if the questionnaire was applied to janitors, I believe the result would be the janitors care more about the health of other janitors. The authors must clarify this point.

7. Finally, I think the authors overlooked some interesting papers related to their research topic, for example:

- https://doi.org/10.1111/jocn.15307

- https://doi.org/10.1007/s40200-020-00643-9

- https://www.medrxiv.org/content/10.1101/2020.08.12.20173476v1

- https://www.medrxiv.org/content/10.1101/2021.05.31.21257245v1

Reviewer #2: Examining social support and professional networks of nurses and nurse technicians during COVID is a timely undertaking, the report of which is the focus of this manuscript. While interesting, there are areas where more information or clarification are needed. These are address below.

General comment: The manuscript would benefit from editorial review, as there are incomplete or confusing sentences throughout the manuscript.

Introduction: Please make clear the gap being addressed by this study as there has been considerable information published about the nurse’s experience during COVID. In addition, 2nd line of 1st paragraph – not certain “control” is the most appropriate word. Would think perhaps “care of infected persons” might be more appropriate.

Research question: Seems focus was on two types of nurse: nurses with baccalaureate and nurse technicians. Suggest rewording RQ. “ What are the social networks of nurses and nurse technicians during COVID?” or something to make it clear you are simply describing social networks of two different nurse types. If you intended to compare, then that should be included in the RQ.

Social Networks: Thank you for including this information. 3rd paragraph in this section: one sentence paragraph – not certain the purpose of this information. Mention is made of SNA studies but no references are provided contributing to a lack of clarity.

Social support at work: Please clarify who provides the assistance and protection to professionals. Perhaps this information could be combined with the previous discussion on social support – with greater emphasis on findings. Also please consider including more than one citation when using phrases such as “several studies.”

Methods: Believe you can use subheadings – which would be very helpful to the reader. Design: suggest using cross-sectional only as that design is one type of observational design. Please include more discussion about the social network analysis methodology. In fact, consider a subheading of data analysis – this would be the place to discuss SNA in greater depth. Describe how the list was developed, based on the literature?? Consider a subheading for participants that provides inclusion criteria, desired number of participants, rationale for two different clinical settings and sites, who approached them to invite them, where were they approached, etc. Were any demographic data collected about the participants? If so please indicate and then describe under findings.

Including a subheading called “data analysis” would strengthen the manuscript and could include the discussion of centrality metrics. Make clear these metrics are the key to SNA. Also include the type of analysis performed – descriptive statistics?

Research Ethics: The statement that is made implies IRB was obtained for a previous study, not this one. Please clarify and if possible, provide the IRB reference number.

Results: Please note phraseology in the first sentence. Should be reordered as something is presented in a table, not a table presenting something – this is a common mistake . However, please consider revising the tables to reflect a clear purpose. Provide sample number- consider only including the top five and bottom five social networks. Somehow, the numbers included in the table need to discussed better. For example, Table 1, Nurse has 0.905 for Degree – but it is not clear what this number means. Is it the mean, total score? Some goes for Table 2. In addition, please be consistent in presenting findings: Nurses first, followed by nurse technicians. Note that differences are references when discussing proximity scores but differences was not mentioned in the research question.

Consider a paragraph at the end of findings that clearly states the conclusions drawn from the findings.

This could be the first paragraph of Discussion instead – what is important is that the reader needs to be clear about what conclusions can be drawn for these data. For example, is it possible nurses and nurse technicians have similar social networks? 3rd paragraph – support the first sentence with citations. Moreover, if social support has been addressed in Brazilian literature, make clear the contribution being made by this study.

Conclusion: The term “informal” is introduced – do not recall seeing the term earlier in the paper. Was this the focus of the study – if so, please make clear from the outset.

Please address limitations associated with the study and make clear the implications. What do findings mean for nurses and technicians?

References: Several references are in a language other the English, making retrieval and understanding of the work inaccessible to an international audience. Please consider replacing with other references.

Figures – please provide legends to help the reader understand information being presented.

Hope comments help.

6. PLOS authors have the option to publish the peer review history of their article (what does this mean?). If published, this will include your full peer review and any attached files.

Reviewer #1: No

Reviewer #2: No

---

## [Author Response · Author response to Decision Letter 0]

30 Mar 2022

Reviewer 1

1. The authors could double check the structural and grammar aspects in the following sentences:

“Networks can, in terms of homogeneity and reciprocity between their actors, be classified as type-1 (total network) or mode-2.”

and

“It’s an observational and cross-sectional research, based on the social network analysis methodology, using an online data

collection instrument.”

*The entire text was submitted to English proofreading.

2. The authors state that

“The study of social networks began in the 1930s within the scope of the social sciences.”

and

“The concept of social support is widely disseminated and studied by the scientific community.”

These statements should be referenced.

*We included the corresponding references.

3. The authors should be clear to all readers, not only those from Social Science. Therefore, I suggest that they introduce the

nomenclature used by social scientists, physicists, and mathematicians in the studying of social networks at the beginning and

adopt only one throughout the manuscript. For example, the authors state that

“The former exists when the actors (nodes) who mention other actors (also called links or alters) are part of the same network, in a

square matrix.”

Links are edges (ties) for Physicists. The authors should avoid this kind of problem.

*We used the most common nomenclature adopted in Social Network Analysis, and attempted to make it more intelligible.

4. It is imperative that the authors include a didactic figure explaining their model precisely. Further, I think the authors could emphasize that their model is a bipartite network at the beginning, not only at the ending.

*We included a didactic frame summarizing the research model (Frame 1).

5. I think the Social Network Analysis performed was quite superficial. There are no histograms or scatter plots of any kind, there is only 1 table and 4 figures (which could be combined into a single panel). Further, the quality of all figures is very low. The authors should improve their analysis.

*This SNA study was based on egocentric networks. Sociograms are usually the only graphic representation employed. 

The four sociograms were organized in one unique figure (Figure 1). Figure met the author’s guide requirements regarding the quality of images.

6. Why are the results not biased due to the chosen sample of participants (only nurses and nursing technicians)? For example, if the questionnaire was applied to janitors, I believe the result would be the janitors care more about the health of other janitors. The authors must clarify this point.

*The research was focused on social relationships between each professional (ego) and other professionals according to their position/category. Each respondent could choose any one of the presented categories. A paragraph was inserted in the final part of the Discussion section, where we comment on this potential bias limitation. 

Reviewer 2

General comment: The manuscript would benefit from editorial review, as there are incomplete or confusing sentences throughout

the manuscript.

Introduction: Please make clear the gap being addressed by this study as there has been considerable information published about the nurse’s experience during COVID. In addition, 2nd line of 1st paragraph – not certain “control” is the most appropriate word.

*We included the lack of studies of this type during the pandemic, and changed the verb "control". 

Research question: Seems focus was on two types of nurse: nurses with baccalaureate and nurse technicians. Suggest rewording

RQ. “ What are the social networks of nurses and nurse technicians during COVID?” or something to make it clear you are simply

describing social networks of two different nurse types. If you intended to compare, then that should be included in the RQ.

*We changed the RQ.

Social Networks: 3rd paragraph in this section: one sentence paragraph – not certain the purpose of this information. Mention is made of SNA studies but no references are provided contributing to a lack of clarity.

*We included the corresponding references.

Social support at work: Please clarify who provides the assistance and protection to professionals. Perhaps this information could

be combined with the previous discussion on social support – with greater emphasis on findings. Also please consider including

more than one citation when using phrases such as “several studies.”

*We merged the two topics (social networks and social support) into one and we corrected the phrases.

Methods: Believe you can use subheadings – which would be very helpful to the reader. Design: suggest using cross-sectional onlyas that design is one type of observational design. Please include more discussion about the social network analysis methodology. In fact, consider a subheading of data analysis – this would be the place to discuss SNA in greater depth. Describe how the list was

developed, based on the literature?? Consider a subheading for participants that provides inclusion criteria, desired number of participants, rationale for two different clinical settings and sites, who approached them to invite them, where were they approached, etc. Were any demographic data collected about the participants? If so please indicate and then describe under findings.

Including a subheading called “data analysis” would strengthen the manuscript and could include the discussion of centrality metrics. Make clear these metrics are the key to SNA. Also include the type of analysis performed – descriptive statistics? Research Ethics: The statement that is made implies IRB was obtained for a previous study, not this one. Please clarify and if

possible, provide the IRB reference number.

*The Methods section was expanded and reorganized. We also added detailed information on the statistical methods. 

Results: Please note phraseology in the first sentence. Should be reordered as something is presented in a table, not a table presenting something – this is a common mistake . However, please consider revising the tables to reflect a clear purpose. Provide sample number- consider only including the top five and bottom five social networks. Somehow, the numbers included in the table

need to discussed better. For example, Table 1, Nurse has 0.905 for Degree – but it is not clear what this number means. Is it the mean, total score? Some goes for Table 2. In addition, please be consistent in presenting findings: Nurses first, followed by nurse technicians. Note that differences are references when discussing proximity scores but differences was not mentioned in the research question. Consider a paragraph at the end of findings that clearly states the conclusions drawn from the findings. This could be the first paragraph of Discussion instead – what is important is that the reader needs to be clear about what conclusions can be drawn for these data. For example, is it possible nurses and nurse technicians have similar social networks? 3rd

paragraph – support the first sentence with citations. Moreover, if social support has been addressed in Brazilian literature, make clear the contribution being made by this study.

*We improved the phraseology and provided the English language proofreading of the manuscript. The centrality measures results are commented both in the Results and Discussion sections.

Conclusion: The term “informal” is introduced – do not recall seeing the term earlier in the paper. Was this the focus of the study – if so, please make clear from the outset.

Please address limitations associated with the study and make clear the implications. What do findings mean for nurses and technicians?

References: Several references are in a language other the English, making retrieval and understanding of the work inaccessible to

an international audience. Please consider replacing with other references.

Figures – please provide legends to help the reader understand information being presented.

*We included a comment about "informal relationships" in the Discussion section. English language was improved and we provided figures’ captions.

---

## [Decision Letter · Decision Letter 1]

27 Apr 2022

PONE-D-21-24493R1Social support and professional networks of nurses and nursing technicians in coping with Covid-19: a sectional study in two Brazilian citiesPLOS ONE

Dear Dr. David,

Thank you for submitting your manuscript to PLOS ONE. After careful consideration, we feel that it has merit but does not fully meet PLOS ONE’s publication criteria as it currently stands. Therefore, we invite you to submit a revised version of the manuscript that addresses the points raised during the review process.

Before the manuscript gets accepted for publication, please answer carefully the remarks raised by the Reviewer #2 related to the item 6, namely, the "Review Comments to the Author" which follows in the report below. 

We look forward to receiving your revised manuscript.

Kind regards,

José S. Andrade Jr.

Academic Editor

PLOS ONE

Journal Requirements:

Reviewers' comments:

Reviewer's Responses to Questions

**Comments to the Author**

1. If the authors have adequately addressed your comments raised in a previous round of review and you feel that this manuscript is now acceptable for publication, you may indicate that here to bypass the “Comments to the Author” section, enter your conflict of interest statement in the “Confidential to Editor” section, and submit your "Accept" recommendation.

Reviewer #1: All comments have been addressed

Reviewer #2: (No Response)

2. Is the manuscript technically sound, and do the data support the conclusions?

Reviewer #1: Yes

Reviewer #2: Yes

3. Has the statistical analysis been performed appropriately and rigorously? 

Reviewer #1: Yes

Reviewer #2: Yes

4. Have the authors made all data underlying the findings in their manuscript fully available?

Reviewer #1: Yes

Reviewer #2: No

5. Is the manuscript presented in an intelligible fashion and written in standard English?

Reviewer #1: Yes

Reviewer #2: Yes

6. Review Comments to the Author

Reviewer #1: (No Response)

Reviewer #2: Authors are to be commended for addressing reviewer concerns. As sometimes happen, when one set of issues is addressed others become apparent. Please clarify the following:

1. Page 6; last paragraph: who provides the social support to professionals?

2. Network data and collection: Information in this section needs to be re-orgranized. Suggest subheadings of Setting and Sample - two separate headings. Combine all the sample information into one section and make clear whether a desired number of participants was needed. Expand on how the sample was recruited - blast email to employees? word of mouth?

3. Instrument: Please provide more information about the questions asked - did anyone review them for validity?

4. Page 10: Human Research Ethics Committee: Information presented here makes this reviewer question whether the current study is secondary analysis of another larger study - please clarify.

7. PLOS authors have the option to publish the peer review history of their article (what does this mean?). If published, this will include your full peer review and any attached files.

Reviewer #1: No

Reviewer #2: No

---

## [Author Response · Author response to Decision Letter 1]

31 May 2022

Response to the Reviewer 2

1. Page 6; last paragraph: who provides the social support to professionals?

R – We provided the requested information.

2. Network data and collection: Information in this section needs to be re-orgranized. Suggest subheadings of Setting and Sample - two separate headings. Combine all the sample information into one section and make clear whether a desired number of participants was needed. Expand on how the sample was recruited - blast email to employees? word of mouth?

R – The section was reorganized. We added the requested information about sample size and recruiting.

3. Instrument: Please provide more information about the questions asked - did anyone review them for validity?

R – We added information about validity.

4. Page 10: Human Research Ethics Committee: Information presented here makes this reviewer question whether the current study is secondary analysis of another larger study - please clarify.

R – We clarified this item.

---

## [Decision Letter · Decision Letter 2]

1 Sep 2022

PONE-D-21-24493R2Social support and professional networks of nurses and nursing technicians in coping with Covid-19: a sectional study in two Brazilian citiesPLOS ONE

Dear Dr. David,

Thank you for submitting your manuscript to PLOS ONE. After careful consideration, we feel that it has merit but does not fully meet PLOS ONE’s publication criteria as it currently stands. Therefore, we invite you to submit a revised version of the manuscript that addresses the points raised during the review process.

We look forward to receiving your revised manuscript.

Kind regards,

Carlos Magno Castelo Branco Fortaleza, M.D., Ph.D.

Academic Editor

PLOS ONE

Journal Requirements:

Additional Editor Comments:

The revised version is improved in clarity, and several questions posed by reviewers were appropriately addressed. However, please pay attention to requests from reviewer #2, concerning information that we believe that are important to improve readability and inform readers appropriately.

Reviewers' comments:

Reviewer's Responses to Questions

**Comments to the Author**

1. If the authors have adequately addressed your comments raised in a previous round of review and you feel that this manuscript is now acceptable for publication, you may indicate that here to bypass the “Comments to the Author” section, enter your conflict of interest statement in the “Confidential to Editor” section, and submit your "Accept" recommendation.

Reviewer #2: All comments have been addressed

Reviewer #3: All comments have been addressed

2. Is the manuscript technically sound, and do the data support the conclusions?

Reviewer #2: Yes

Reviewer #3: Yes

3. Has the statistical analysis been performed appropriately and rigorously? 

Reviewer #2: Yes

Reviewer #3: Yes

4. Have the authors made all data underlying the findings in their manuscript fully available?

Reviewer #2: Yes

Reviewer #3: Yes

5. Is the manuscript presented in an intelligible fashion and written in standard English?

Reviewer #2: Yes

Reviewer #3: Yes

6. Review Comments to the Author

Reviewer #2: (No Response)

Reviewer #3: 1. In my opinion this st study presents results of original research, in special the theme nursing and COVID-19.

2. The method and data analyses performed to a high technical standard and are described in sufficient details.

4. Conclusions are presented in an appropriate fashion and are supported by the data.

5. The ethics procedures were described.

7. The article adheres to appropriate reporting guidelines and community standards for data availability.

7. PLOS authors have the option to publish the peer review history of their article (what does this mean?). If published, this will include your full peer review and any attached files.

Reviewer #2: No

Reviewer #3: **Yes: **Rosely Moralez de Figueiredo

---

## [Author Response · Author response to Decision Letter 2]

23 Nov 2022

We authors have already responded to reviewers twice. Please, refer to anterior files.

---

## [Editor Report · Decision Letter 3]

28 Dec 2022

Social support and professional networks of nurses and nursing technicians in coping with Covid-19: a sectional study in two Brazilian cities

PONE-D-21-24493R3

Dear Dr. David,

We’re pleased to inform you that your manuscript has been judged scientifically suitable for publication and will be formally accepted for publication once it meets all outstanding technical requirements.

Kind regards,

Carlos Magno Castelo Branco Fortaleza, M.D., Ph.D.

Academic Editor

PLOS ONE

Additional Editor Comments (optional):

The authors provided appropriate answers to the reviewers' requirements and changed the manuscript appropriately. The version that was reinserted in the editorial manager does provide answers to the questions posed by reviewers.
---

## [Editor Report · Acceptance letter]

11 Jan 2023

PONE-D-21-24493R3 

Social support and professional networks of nurses and nursing technicians in coping with Covid-19: a sectional study in two Brazilian cities 

Dear Dr. David:

I'm pleased to inform you that your manuscript has been deemed suitable for publication in PLOS ONE. Congratulations! Your manuscript is now with our production department. 

Kind regards, 

on behalf of

Dr. Carlos Magno Castelo Branco Fortaleza 

Academic Editor

PLOS ONE